# C. elegans collectively forms dynamical networks

Takuma Sugi [ID] [1], Hiroshi Ito[2], Masaki Nishimura[1] & Ken H. Nagai[3]

Understanding physical rules underlying collective motions requires perturbation of controllable parameters in self-propelled particles. However, controlling parameters in animals is generally not easy, which makes collective behaviours of animals elusive. Here, we report an experimental system in which a conventional model animal, *Caenorhabditis elegans*, collectively forms dynamical networks of bundle-shaped aggregates. We investigate the dependence of our experimental system on various extrinsic parameters (material of substrate, ambient humidity and density of worms). Taking advantage of well-established *C. elegans* genetics, we also control intrinsic parameters (genetically determined motility) by mutations and by forced neural activation via optogenetics. Furthermore, we develop a minimal agent-based model that reproduces the dynamical network formation and its dependence on the parameters, suggesting that the key factors are alignment of worms after collision and smooth turning. Our findings imply that the concepts of active matter physics may help us to understand biological functions of animal groups.

[1] Molecular Neuroscience Research Center, Shiga University of Medical Science, Otsu, Shiga 520-2192, Japan. [2] Faculty of Design, Kyushu University, Fukuoka 815-8540, Japan. [3] School of Materials Science, Japan Advanced Institute of Science and Technology, Nomi, Ishikawa 923-1292, Japan. Correspondence and requests for materials should be addressed to T.S. (email: tsugi@belle.shiga-med.ac.jp) or to H.I. (email: hito@design.kyushu-u.ac.jp) or to K.H.N. (email: k-nagai@jasit.ac.jp)

Ordered collective motions are ubiquitous among locally interacting living beings[1–3] and play significant roles in biological functions such as biofilm formation[4], wound healing[5], the flocking of birds, and the schooling of fish[6–8]. Research on the physics of active matter, which seeks to identify unified descriptions of such collective motions, has included various studies using mathematical models with only simple rules of self-propulsion[3,9,10]. Experimental systems have been developed for non-living self-propelled particles (SPPs)[11–13], bacteria[4,14], and mammalian cells on a substrate[15,16], but there have been no experimental systems of multicellular organisms, which have much more complex behaviours[17,18], with various controllable parameters over a wide range[19]. Thus, whether the collective motions of animals can be described by minimal models remains unknown.

Some species of nematodes, particularly parasitic nematodes, have long been known to aggregate and swarm on their habitats to survive desiccation for extended periods[20,21]. This swarming was quantitatively proven to be one of the most important survival strategies for adaptation to the demands of fluctuating environments that occasionally become extreme and life-threatening[22–24]. The free-living nematode *Caenorhabditis elegans*, a commonly used laboratory model animal, is genetically tractable and thus potentially offers a great opportunity to experimentally investigate the collective motions of animals under the control of a wide variety of parameters. However, *C. elegans* researchers have paid little attention to establishing an experimental system for producing large-scale *C. elegans* swarms[21] because it is difficult to prepare very large numbers of worms on a solid surface in the standard cultivation protocol of *C. elegans* due to the decrease in its propagation speed as food becomes limited.

In this study, we show that *C. elegans* collectively forms dynamical networks of bundle-shaped aggregates with simple physical rules. We apply a culture method previously established for other nematodes, in which dog food agar (DFA) medium is used[25,26]. Using the established experimental system, we examine the dependence of this system on extrinsic parameters (the substrate, ambient humidity, and density of worms) and intrinsic parameters (genetically determined motility) by mutations or optogenetics. We also construct an agent-based model with simple rules that reproduces the dynamical network and its dependence on the parameters, suggesting that the key rules are alignment and attraction of worms after collision and smooth turning. Our findings imply that the concepts of active matter physics may contribute to the understanding of biological functions of animal groups.

## Results

**Highly propagated *C. elegans* forms dynamical network**. DFA medium contains enough nutrients to enable the propagation of a very large number of worms for a long time (>1 month). During propagation, the worms are kept in the dauer state, a starvation-like state induced by crowded conditions[25–27]. In a week of cultivation in DFA medium in a glass bottle, the propagated *C. elegans* sometimes climbed up the inner wall of the bottle. We noticed that when the bottom of a bottle was warmed from 23 to 25 °C, large numbers of propagated worms climbed up the inner surface of the bottle. We then observed the emergence of a dynamical network structure on the inner surface of the glass bottles (Fig. 1a, b and Supplementary Movie 1). The structure comprised a large number of compartments surrounded by bundle-shaped worm aggregates. The average diameter of the

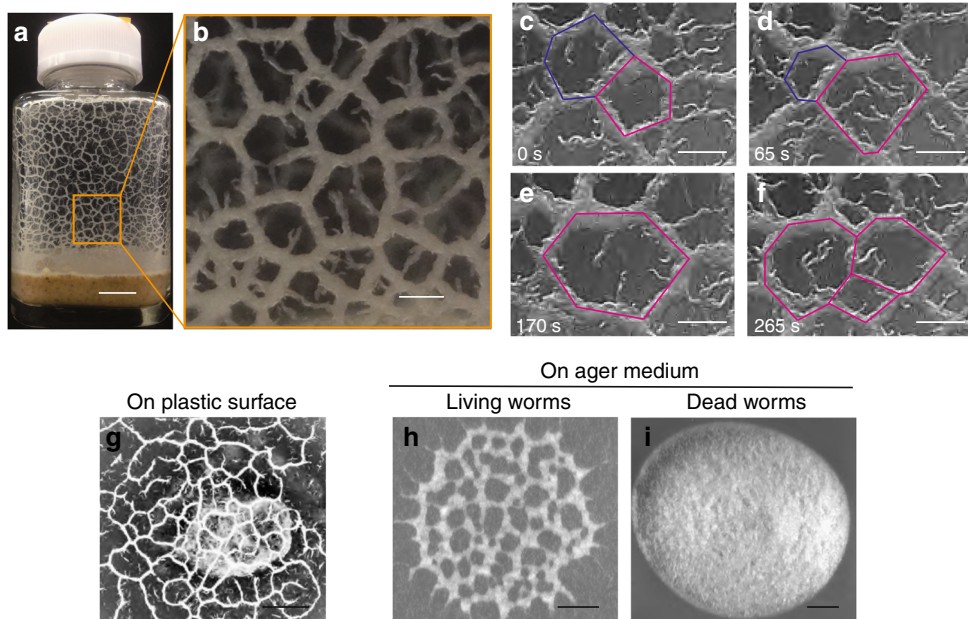

**Fig. 1** Dynamical network formed by collective nematode behaviour. **a** Network pattern generated by *C. elegans* population cultured in a glass bottle. The network was formed on the lateral surface of the glass bottle. Scale bar, 10 mm. **b** A magnified image of **a**. Scale bar, 2 mm. **c–f** Dynamical network pattern. The shapes of compartments were not fixed. Scale bar, 1 mm. **g** Network formation on a plastic substrate. Scale bar, 4 mm. **h, i** The network was formed on agar medium by living worms (**h**) but not by dead worms (**i**). The initial worm concentration was 1500 worms μl$^{-1}$ (300,000 worms in 200 μl of water). Dead worms were prepared by incubating worm suspensions at 70 °C for 10 min before the assay. Since living worms actively dispersed as the water seeped into the medium, the number of worms forming the network structure in **h** is smaller than the number of dead worms in **i**. Scale bar, 4 mm

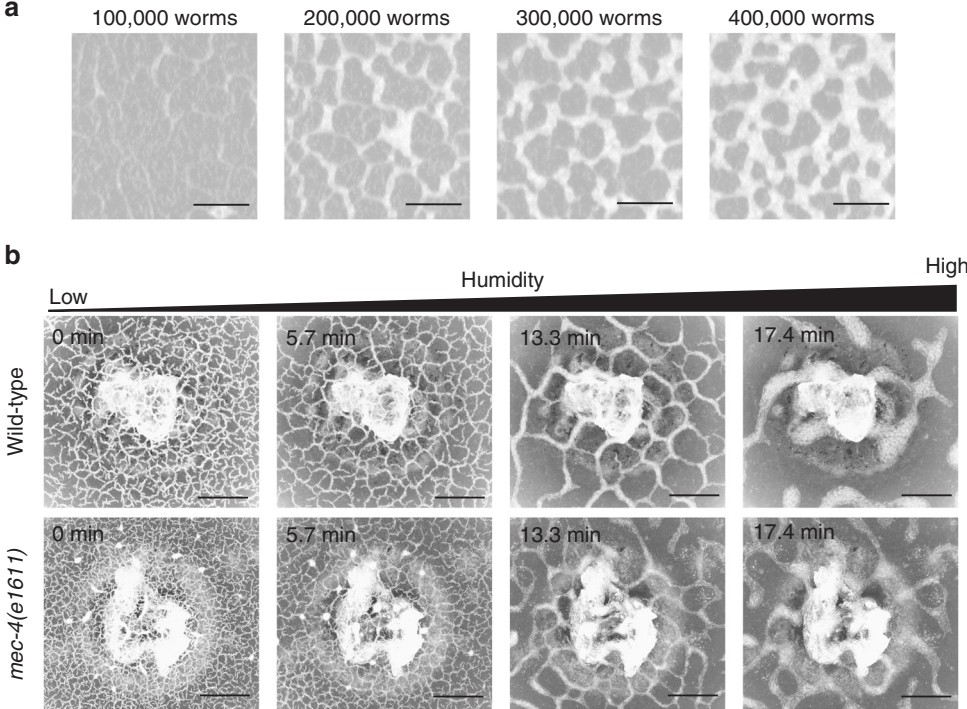

**Fig. 2** Dependence of the network pattern on extrinsic parameters. **a** Density dependence of *C. elegans*. The number of worms on each agar surface was from 100,000 to 400,000 worms. Scale bar, 4 mm. **b** Dependence of the *C. elegans* network on the ambient humidity and movement curvature of isolated single worms. The *mec-4(e1611)* mutant (lower panel), which moves with higher average curvature than the wild-type animal (upper panel), was used to vary the movement curvature. The average size of the compartments was 0.200 mm$^2$ at 0 min, 0.482 mm$^2$ at 5.7 min, and 8.878 mm$^2$ at 13.3 min in the upper row and was 0.090 mm$^2$ at 0 min, 0.241 mm$^2$ at 5.7 min, and 3.814 mm$^2$ at 13.3 min in the lower row. The object at the centre is a lump of dog food on the agar, not on the lid. Scale bar, 4 mm

compartments was clearly larger than the typical body size of a single worm (431 ± 64 μm (s.d., $n = 13$)). The bundle in the magnified image in Fig. 1b was highly crowded with worms, indicating that the effect of excluded volume was significant. The network pattern dynamically remodelled over time by the repeated coalescence and division of compartments in approximately 100 s (Fig. 1c–f).

We find that the worms can form a qualitatively identical network on a plastic substrate by climbing up to the inner surface of the lid from DFA on agar medium inside a Petri plate (Fig. 1g, Supplementary Figure 1). We also examined whether the network could be formed on a food (*Escherichia coli* OP50)-free agar surface of nematode growth medium (NGM), which was used in the standard cultivation method instead of the naturally habitable soil environment[28]. For this experiment, we collected large numbers of worms that were propagated in DFA medium and climbed up to the inner surface of the lid using water, and then the obtained droplet was placed on an agar surface. We observed that the worms also formed a dynamical network after water seeped into the agar medium (Fig. 1h). The qualitatively identical collective motions on three different substrates indicate that *C. elegans* forms this dynamical network regardless of the substrate material as long as the number of worms is large enough. The network formation is noted to require self-propelled activity, as no obvious self-organized pattern was observed for dead worms on agar (Fig. 1i).

**Formation of dynamical network depends on both extrinsic and intrinsic parameters.** This reconstructed system of living animals enables us to easily control certain extrinsic parameters, such as the density of worms and the humidity in a

Petri plate. To control the density, we collected propagated worms from the inner surfaces of lids and prepared several 200 μl droplets with worm densities in the range of 500–2000 worms μl$^{-1}$. Each of the droplets was placed on the agar surface. We found that the formation of the dynamical network required a worm density greater than 1000 worms μl$^{-1}$ (200,000 worms on the agar surface) (Fig. 2a). We also controlled the humidity as another extrinsic parameter. To increase the humidity around the worms, a plate with agar in the bottom and worms on the lid was heated from 23 to 26 °C from the bottom at a speed of ca. 1 °C per min (Supplementary Figure 1). The heating evaporated the water in the agar to the air inside the Petri plate, thereby gradually increasing the humidity. As previously reported for another nematode, *Panagrellus redivivus*[29], we confirmed that the attachment of colliding worms was driven by the surface tension of the water around the worms, and this attractive force was strong enough to keep the worms in an aggregated state in opposition to the repulsive force exerted by their muscles. The increase in humidity strengthened the attraction (Supplementary Movie 2). The time evolution of the network on the lid is shown in Fig. 2b and Supplementary Movie 3. A clear dynamical network was observed at the beginning of the experiment (the upper row of Fig. 2b (0 min)). Then, as the humidity increased, the compartments grew larger (the upper row of Fig. 2b (5.7 and 13.3 min) and Supplementary Movie 3). Further increasing the humidity led to the collapse of the network structure and the formation of many simple aggregates (the upper row of Fig. 2b (17.4 min)). For a while, the shape of the aggregates largely fluctuated and small running aggregates sometimes split from protuberances at the edges of the aggregates (Supplementary Movie 4). Finally, all the aggregates became still.

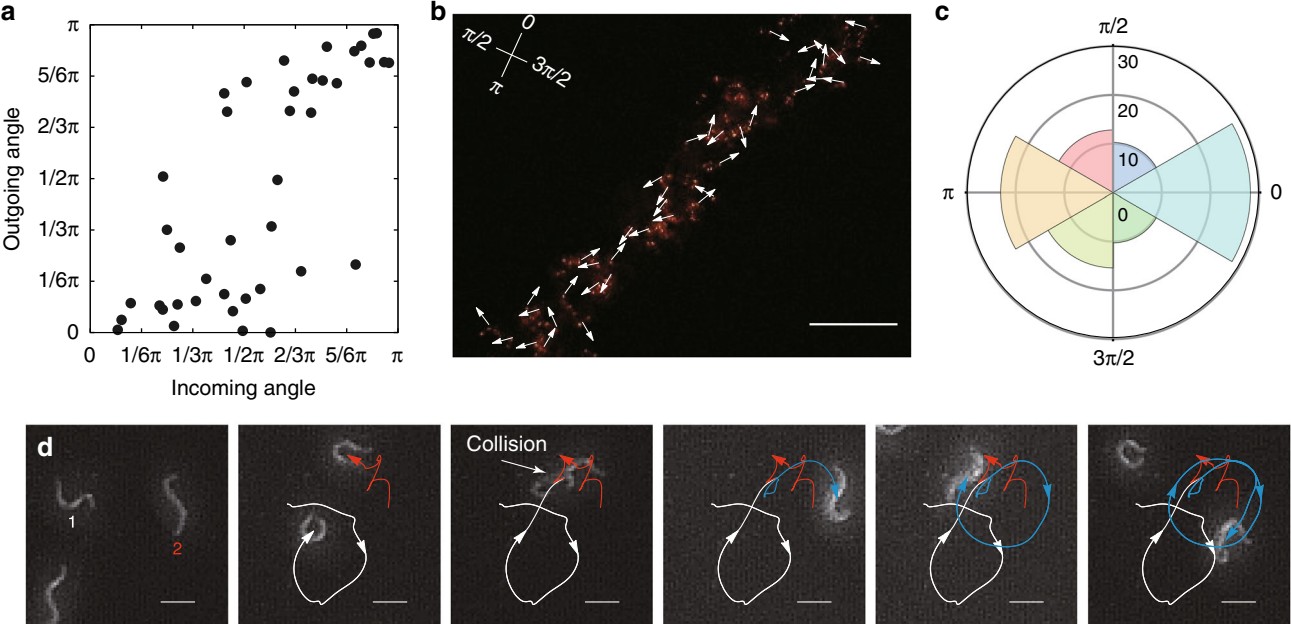

**Fig. 3** Nematic interactions between worms. **a** Incoming–outgoing angles of collisions between worms. Forty-three collisions were analysed. **b** Direction of each worm's movement in a bundle. The white arrows indicate the movement directions of worms that moved more than 80 μm (39 worms) in 10 s on the inner surface of Petri plate lid. Each worm's position was identified by manually tracking the fluorescence of TagRFP expressed in their few head neurons. Because the gene encoding TagRFP was introduced into the worms as an extrachromosomal array, approximately 70% of the dauer worms express this protein. Scale bar, 10 mm. **c** Pie chart of worm movement directions in **b**. Angle 0 was defined based on the averaged movement direction of 39 worms. **d** Representative trajectories of moving worms. The trajectories for two worms that moved separately for the initial 105 s are indicated in white and red. The trajectory of the two worms that collided at 105 s and then moved together for at least 620 s is indicated in blue. Scale bar, 200 μm

The accumulated genetic resources and tools, especially related to motility, that have been developed in *C. elegans*[30] enable easy control of the intrinsic parameters of the collective motion of the worms. We examined the collective motion of the *mec-4(e1611)* mutant, which carries a mutation in the gene encoding the mechanosensory Na+ channel MEC-4 and moves along a circular trajectory with higher curvature than that of the wild-type worm[31]. We cultivated the *mec-4(e1611)* mutant in DFA on NGM inside a Petri plate and increased the humidity inside the plate, as we did for the wild-type worms (the lower row of Fig. 2b and Supplementary Movie 5). Subsequently, we observed similar network formation by the *mec-4(e1611)* mutants during changes in humidity, and then its compartment size was smaller than that of the wild-type worms. The smaller compartment sizes were also confirmed in another *mec-4* allele, *mec-4(e1339)*[32] (Supplementary Figure 2), supporting that loss of MEC-4 function is responsible for the observed phenotype. We further tried to examine the other mutants, *snt-1(md290)* mutant[33] and *let-60(lf)* mutant[34], both of which have been known to move with higher curvature, but these mutants did not show enough propagation in DFA. We therefore cannot exclude the possibility that the mutations have effects on characteristics of motility other than curvature, such as mechanosensory response, and thereby affects the compartment size of the network. We next constructed an experimental system for optical manipulation of the network. This newly established optogenetic system (see Methods) enables a transient perturbation to the worm aggregations by the optical activation of mechanosensory neurons. Blue light illumination has been known to competitively drive reversal, accelerate forward movements, and induce the movement of halted worms (approximately 20% of worms before light illumination). As soon as the light was turned on, the bundle started to collapse. We find that the bundle can recover to the original shape as long as the duration of activation was not long

(<2 s) (Supplementary Movie 6). Activation with mild light intensity for a prolonged duration (>30 s) broke the bundle. After the light was turned off, the worms formed different bundles (Supplementary Movie 7). It is noted that without all-trans-retinal (ATR), which is needed for the optogenetic activation of the target neuron, no response of the bundles to light illumination was observed (Supplementary Movie 8).

**Worms align in nematic order through collisions.** To elucidate the key behavioural characteristics necessary for network formation, behavioural data are collected at the single-worm level. We analysed the pair interactions and trajectories of sparsely isolated worms (~1 worm mm$^{-2}$) on the inner plastic surface of the lid of an NGM plate. A large number of random collisions occurred between worms in close proximity. In the 43 pair collision events, the outgoing angles were near 0 or $\pi$, irrespective of the value of the incoming angles (Fig. 3a). This result indicates that collisions induce the alignment or anti-alignment of worms, i.e. nematic rather than polar order. We find that individual *C. elegans* in the network are also nematically aligned by using transparent worms labelled with a fluorescent protein TagRFP. A total of 51.3% of worms moved in the directions $0 \pm \pi/12$ rad (28.2%) and $\pi \pm \pi/12$ rad (23.1%) along a bundle (Fig. 3b, c). The isolated dauer worms exhibited clockwise or anti-clockwise circular trajectories with a gradual change in rotation rate, as observed for fed worms in a previous study[31] (Fig. 3d and Supplementary Movie 9). These trajectories included short-wavelength oscillation, which arises from the worms' undulatory locomotion. From the analysis of 38 worms (see Methods), we obtained the mean velocity ($v_0 = 87 \pm 30$ (s.d.) μm s$^{-1}$) and the mean rotation rate ($\omega_0 = 1.4 \times 10^{-3} \pm 1.9 \times 10^{-2}$ (s.e.) rad s$^{-1}$). The standard

deviation and the correlation time of the rotation rate were estimated to be $\sigma_\omega = 0.155$ (95% confidence interval, 0.151–0.159) rad s$^{-1}$ and $\tau = 27$ (95% confidence interval, 25–29) s, respectively.

**Mathematical model reproduces dynamical network.** The short-range nematic alignment and smooth turning of *C. elegans* are reminiscent of microtubules driven by axonemal dynein c. The hexagonal lattice of vortices formed by the microtubules was reproduced by a simple agent-based model, in which the agents have a memory of the rotation rate[11]. To confirm whether the collective motion of *C. elegans* could be reproduced by the minimal model, we employ one based on the model in refs. [11,35]. In addition to short-range nematic interaction and smooth turning, two additional significant characteristics, the attraction caused by surface tension ($\mathbf{F}^a$) and the repulsion due to the excluded volume of the nematodes ($\mathbf{F}^r$), are considered. The simulated equations are

$$\dot{\mathbf{r}}_i = \mathbf{e}_{\theta_i} + \sum_{r_{ij} < r^r} \mathbf{F}^r_{ij} + \sum_{r^r < r_{ij} < 1} \mathbf{F}^a_{ij} \qquad (1)$$

$$\dot{\theta}_i = \omega_i + \frac{1}{N_i} \sum_{r^r < r_{ij} < 1} \sin 2\left(\theta_j - \theta_i\right) \qquad (2)$$

$$\dot{\omega}_i = -\frac{\omega_i - \omega_0}{\tau} + \sqrt{\frac{2}{\tau}} \sigma_\omega \xi_i \qquad (3)$$

$$\mathbf{F}^r_{ij} = k^r \left(r_{ij} - r^r\right) \mathbf{e}_{ij} \qquad (4)$$

$$\mathbf{F}^a_{ij} = \frac{k^a}{r_{ij}} \mathbf{e}_{ij}, \qquad (5)$$

where $\mathbf{r}_i$, $\theta_i$, and $\omega_i$ are the position, the direction of motion, and the rotation rate of particle $i$, respectively, and $\mathbf{e}_{\theta_i}$ is the unit vector in the direction of $\theta_i$, which means that isolated particles move with the speed of 1. The repulsive force, $\mathbf{F}^r_{ij}$, is exerted on particle $i$ by particle $j$ in a circle with a radius of $r^r$ and its centre at $\mathbf{r}_i$. The direction of $\mathbf{F}^r_{ij}$ was the same as the unit vector from particle $i$ to particle $j$, $\mathbf{e}_{ij}$. When $r^r < r_{ij} = |\mathbf{r}_i - \mathbf{r}_j| < 1$, an attractive force, $\mathbf{F}^a_{ij}$, is exerted by particle $j$. The inverse of the distance of the particles is chosen as the dependence of $\mathbf{F}^a_{ij}$ on the distance from the neighbours, with reference to ref. [36]. In the area $\mathbf{F}^a_{ij}$ works, particle $i$ aligns head to head or head to tail with particle $j$. The alignment interaction term is normalized by the number of interacting particles to avoid excessively strong interactions. The Euler method with time interval of 0.001 is used to solve the equations numerically. The values of $r^r$ and $k^r$ are fixed to 0.2, and 10, respectively.

Since the particle's speed is 1 and the radius of the interaction range is 1, the parameters corresponding to the experiments are the estimates from single-worm tracking normalized by half the body length and the average time during movement over half the body length. Half the body length is chosen because the worms are shortened due to the undulation of their bodies during crawling. When the average ($\omega_0 = 0.0035$) and standard deviation ($\sigma_\omega = 0.35$) of the rotation rate and the rotational correlation time ($\tau = 10$) corresponding to the aforementioned estimated values are used, a dynamical network is successfully formed, as shown in Fig. 4a and Supplementary Movie 10.

To compare the dynamical networks of the model and the worms, we measure the area surrounded by thick bundles (see

Methods). In the large size range, the histogram of the logarithm of size takes the form of a Gaussian distribution in the model case (Fig. 4b). Indeed, the quantile–quantile plot[37] of the fitted Gaussian curve vs. the histogram in the range larger than 50 was linear (inset in Fig. 4b). In the case of the dynamical network of *C. elegans* in Supplementary Movie 1, in the size range larger than 0.73 mm$^2$, the histogram of the logarithm of compartment size also took the form of a Gaussian distribution, as shown in Fig. 4c. The same distribution form in the real and the model case indicates that both dynamical networks are governed by a common physical process and that the minimal model reproduced the dynamical network formation of the worms. The log-normal distribution of object size is often observed in random nucleation and growth processes[38] and in pulverization[39]. When objects are divided into fragments, the division in which the ratio of the original objects and the fragments is distributed independently of the original particle size leads asymptotically to a log-normal size distribution. Thus, the log-normal distribution in the dynamical networks suggests that the compartments divide independently of the area size.

**Minimal mathematical model reproduces extrinsic and intrinsic parameter dependence.** By the model, the extrinsic parameter dependence of the network is well reproduced. A sufficiently large density of worms was needed in the model, as in the case of real worms (Figs. 2a and 4d). Fig. 4e shows the $k^a$ dependence, which corresponds to the humidity dependence of the network of worms. The upper figures of Fig. 4e is in the case of the model with parameters estimated from single-worm tracking. The size of the compartments gradually increased with increasing $k^a$, and non-motile aggregates were formed when $k^a$ was larger than the critical value, as in the case with increasing humidity. It should be noted that the compartment size of real worms network more strongly depends on humidity than results from our model.

The intrinsic parameter dependence was also compared. As stated above, the *mec-4* mutant formed a dynamical network with smaller compartments (Fig. 2b). In the simulations, the particles travelling with higher $\sigma_\omega$ and $\omega_0$ formed the smaller compartments (Fig. 4e). As for the optical stimulation, our model responded to the increase in the number of active particles in a similar manner to the optogenetically induced response. When 30% of the particles were inactive, the dynamical network was formed, as in the initial state of Supplementary Movie 11. To make the particles inactive, zero is multiplied to the terms $\mathbf{e}_{\theta_i}$ in the first equation and $\omega_i$ in the second equation. The sudden change from the behaviour of the inactive particles to that of normal particles collapsed the network structure. Once the ratio of inactive particles returned to its initial value, the network reformed. Although light stimulation leads to both reversal and forward acceleration of worms, at least we can conclude that the decrease of halted worms is one of the main causes of the instability of the bundles.

## Discussion

In this study, we construct an animal model system to study the large-scale collective behaviour in the laboratory (Fig. 1a). We further develop an agent-based model that reproduces the dynamical network formation and its dependences on the experimentally controlled parameters, demonstrating that even the collective motion of *C. elegans* can be described by a simple minimal model in a broad area of parameter space. Thus, the behavioural rule underlying a newly found biological phenomenon is understood from the perspective of active matter physics.

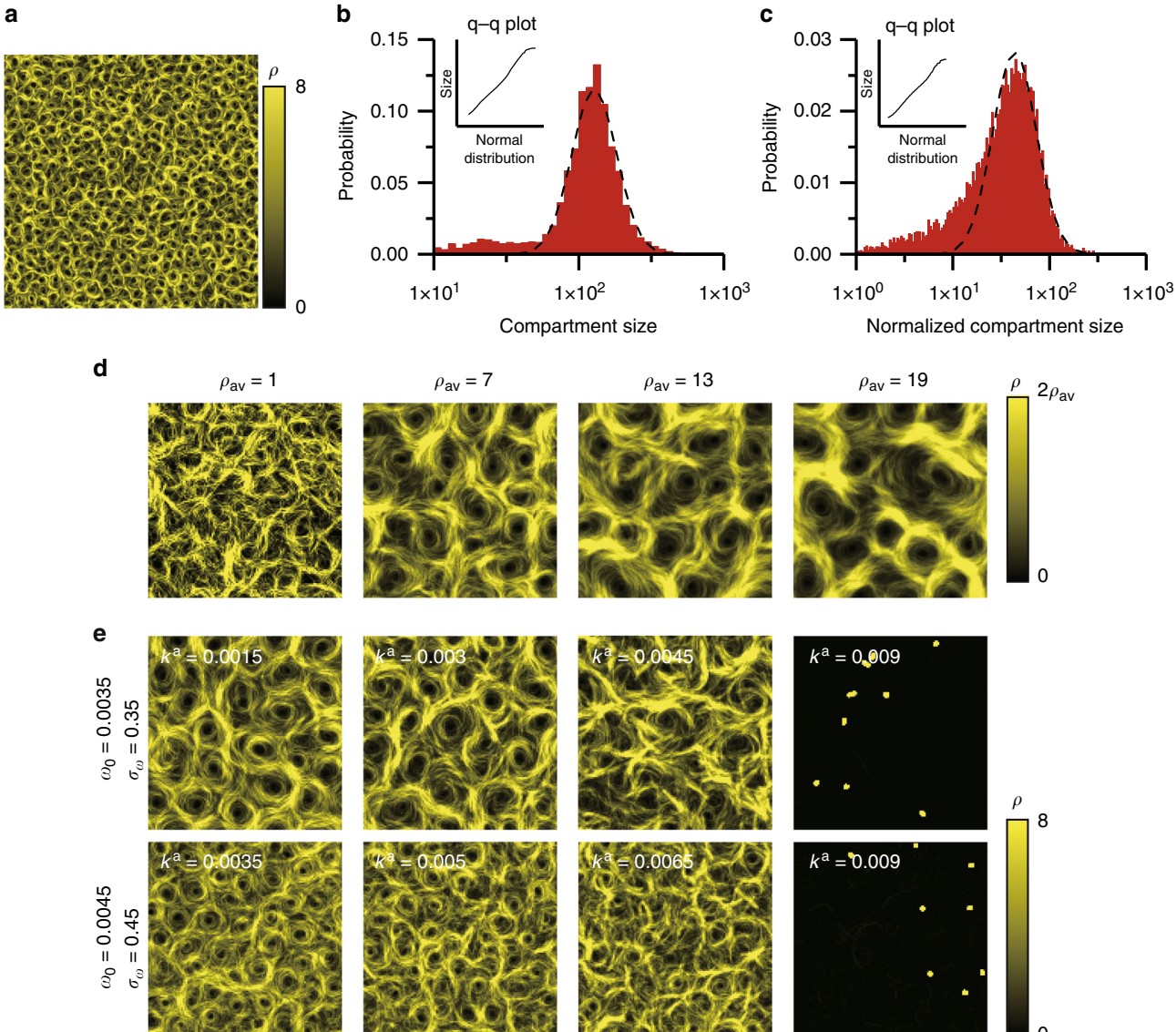

**Fig. 4** Mathematical model for the dynamical network of *C. elegans*. The colour shows the average local density over 10 time units. The images were constructed from the data of more than 1000 time units from the random initial conditions. The area of simulation was 256 unit length square in **a** and 64 unit length square in **b**–**e**, respectively. The boundary condition was periodic. **a** Dynamical network of the model. The average numerical density was 4 per unit length square, $k^a = 0.002$. **b**, **c** Distribution of the size of compartments in the dynamical network from the model (**b**) and the experiment (**c**). The size was normalized by the square of half the body length in **c**. The shape of the distribution was a log-normal distribution in the large size range. The inset shows the quantile–quantile plot of the fitted Gaussian curve vs. the histogram in the large size range. **d** Dependence on average particle density, $\rho_{av}$. $k^a = 0.002$. **e** Collective motion dependence on $k^a$, $\omega_0$, and $\sigma_\omega$. The average numerical density was 4 per unit length square. In the upper row, $\omega_0 = 0.0035$, $\sigma_\omega = 0.35$, in the lower row, $\omega_0 = 0.0045$, $\sigma_\omega = 0.45$. The average size of compartments larger than 30 unit length square was 133 unit length square ($k^a = 0.0015$), 135 unit length square ($k^a = 0.003$), and 167 unit length square ($k^a = 0.0045$) in the upper row and 57 unit length square ($k^a = 0.0035$), 63 unit length square ($k^a = 0.005$), and 68 unit length square ($k^a = 0.0065$) in the lower row. Small spurious compartments inside bundles were detected because point particles were used; therefore, the small size range was neglected in averaging the size

Active matter physics research seeks to identify unified descriptions of collective motions of non-living and living SPPs. For this goal, so far, experimental systems have been developed for non-living SPPs[11–13] and cells[4,14–16], but there have been no systems developed for multicellular organisms[19], which have much more complex behaviours[17,18]. Therefore. our finding of such a new phenomenon extends the possibility for the existence of a unified description of collective motions.

The nematode *C. elegans* was introduced as a genetically tractable laboratory animal[28,30]. Since then, behavioural genetic studies using this model animal have contributed to the investigation of individual-level behavioural paradigms. However, over the past 50 years, although a simple clumping pattern was observed[40,41], no reports have demonstrated dynamical pattern formation via the group-level behaviour of *C. elegans*. One of the breakthroughs in this study is the use of DFA medium, which allows the easy maintenance of a very large number of worms for an extended time in a Petri plate. Using DFA medium, we have presented the first observation of dynamical collective behaviour by *C. elegans*, thereby

introducing a new behavioural paradigm. This finding implies that if a very large number of animals can be maintained, we might find unknown dynamical behaviours much larger than their bodies even in traditionally used model animals, such as larval zebrafish. The great advantage of this experimental system using *C. elegans* is the easy simultaneous observation of the dynamics at both the individual and collective levels thanks to its submillimetre body length. This advantage and the genetic tractability of *C. elegans* offer unprecedented opportunities to control not only extrinsic but also intrinsic parameters, enabling examination of whether the collective behaviours can be described by a simple model. Although there are some inconsistencies between the worm's behaviours and the model, e.g. compartment size dependence on humidity, we can state that our results provide a simple physical description of animal collective behaviours. In this study, individual trajectories in the network, and collapsing process and effects of reversal of worms under the optical stimulation are not investigated well. The careful investigation of these subjects will lead to further understanding of the collective motion of *C. elegans*. As aggregation is the strategy by which nematodes resist desiccation or forage[22,40,41], we expect that network formation can also be linked to survival strategies. The physiological basis of nematode survival should be reconsidered with the help of the physics of active matter.

## Methods

***C. elegans* strain preparation**. The wild-type N2 Bristol strain, the *mec-4(e1611)*[32], *mec-4(e1339)*[32], *snt-1(md290)*[33], and *let-60(lf)*[34] mutant strains and ZX899 strain (*lite-1(ce314); ljIs123[mec-4p::ChR2, unc-122p::RFP]*)[42], in which the extra-chromosomal array *zxEx621* from the original ZX899 strain were removed, were used in this study. For single-worm tracking, the AVA neuron of *C. elegans* was marked by expressing fluorescent protein TagRFP under the control of the *flp-18* promoter[43] according to the standard protocol of germline transformation[44]. These strains were maintained and handled by standard methods[28].

**Cultivation with DFA medium in glass bottle**. Large numbers of *C. elegans* dauer larvae were obtained by cultivating worms in DFA medium[25,26]. A glass sample bottle (Toseiyoki) containing 2 g of powdered dog food (VITA-ONE, Nihon Pet Food) and 5 ml of 1% agar medium was autoclaved and cooled to room temperature. Then, several drops of *E. coli* OP50 suspension in Luria–Bertani broth were added to the surface of the medium. To prepare the worms for inoculation into the DFA medium, four well-fed wild-type adult worms were initially deposited on a 60 mm plate (Thermo Scientific) containing 14 ml of NGM with agar, on which *E. coli* OP50 was seeded[28]. F1 worms were grown to starvation in the NGM plate at 23 °C for 4 days. The starved worms were collected from four NGM plates using autoclaved water and then inoculated into DFA medium in a glass bottle, which was then incubated at 23 °C for 7–10 days. The temperature of the bottom surface was changed from 23 to 25 °C using a Peltier temperature controller unit (Ampere). The network structures on the inner surfaces of the two glass bottles were recorded with USB-controlled CCD cameras (Sentech), which were each coupled to a 25 mm focal-length and C-mount machine vision lens (Azure) and a C-mount adapter (thickness of 5 mm). The frame rate was less than 0.20 frames s$^{-1}$.

**Cultivation with DFA on NGM plate and attraction strength control**. Before inoculation with worms, small amounts (approximately 0.5 g) of DFA medium were autoclaved in a glass beaker and transferred onto the centre of an NGM plate on which *E. coli* OP50 was seeded. Then, starved worms collected from two NGM plates were inoculated onto the DFA on the NGM plate. Worms propagated at 23 °C climbed up to the lid of the plate for approximately 10 days.

On the day of the experiment, a Petri plate was placed onto an aluminium plate on the stage of an Olympus SZX7 microscope, which was kept at 23 °C by a Peltier temperature controller unit (Vics). The Petri plate was maintained under this condition for 5 min before the image acquisition. Then, the temperature of the bottom of the Petri plate was increased from 23 to 26 °C to change the humidity inside the plate. Image acquisition of the inner surface of the plate lid was performed by the DP74 colour camera (Olympus) at 1.0 frame s$^{-1}$. A Plan Achromat objective lens (×0.5, NA = 0.05; Olympus) was used as a low-magnification objective. The acquired images were saved in the Tagged Image File format.

**Reconstruction of *C. elegans* network on agar surface**. Worms were propagated with DFA on NGM in a Petri plate. The worms that moved to the lid of the plate were collected with autoclaved water and washed once. The concentration of purified worms in water was estimated by counting the number of worms in a part of the worm suspension. Based on this estimation, several worm suspensions were prepared with varying concentrations in the range of 500 worms μl$^{-1}$ (100,000 worms in 200 μl of water) to 2000 worms μl$^{-1}$ (400,000 worms in 200 μl of water). The worm suspension was dropped onto a surface of food-free NGM in a Petri plate. Images of the agar surfaces were captured by a WSE-6100H LuminoGraph image analyser (ATTO).

**Single worm tracking in a bundle**. The position of a worm in a bundle was often impossible to measure because the worms in the bundle were too crowded. Therefore, a few head neurons of worms were marked by expressing the fluorescent protein TagRFP (Fig. 3b). The sparse TagRFP expression made it easy to locate individual worms. Images of worms on a plastic surface were taken by a Leica MZ10F fluorescence microscope equipped with a Plan apochromatic objective lens (×1.0, NA = 0.125; Leica Microsystems) at 15 frames s$^{-1}$. The movement directions of 39 worms in a bundle for 10 s were manually identified based on the positions of the fluorescence. The pie chart in Fig. 3c was created by analysing all movement directions with Mathematica 9.0 (Wolfram).

**Optogenetics**. Optogenetic experiments were performed with a blue light-activated cation channel, channelrhodopsin-2 (ChR2), which has been used to noninvasively control the activity of well-defined neuronal populations[45,46]. ZX899 is the light-insensitive mutant *lite-1(ce314)* that carries ChR2 under the control of the *mec-4* promoter[42,47]. ChR2 is expressed in the six touch neurons, and the activation of these neurons by blue light illumination has been known to competitively drive reversal and accelerate forward movements as major and minor responses, respectively[42]. Blue light illumination not only accelerated the movement of worms (approximately 78.9%) that were already moving before illumination but also stimulated all inactive halted worms (approximately 21.1%) to move actively. All worms harboured ChR2 because the extrachromosomal array including the *mec-4p::ChR2* DNA was integrated into the chromosomes[42]. Worms were cultivated at 23 °C with DFA on NGM in a Petri plate under dark conditions. 40 μl of 50 μM ATR (Sigma-Aldrich), the cofactor of ChR2, was poured onto the DFA before cultivation.

Worms climbing onto the lid of a Petri plate were illuminated on the stage of an Olympus SZX7 microscope, which was maintained at 23 °C. The Petri plate was maintained under these conditions for 5 min before blue light illumination. For ChR2 activation, a 100 W mercury lamp (Olympus) was used to deliver blue light, filtered with an SZX-MGFP set (Olympus). The illumination time was precisely controlled using a BSH2-RIX-2-1 electromagnetic shutter system (SIGMAKOKI). Image acquisition of the inner surface of the plate lid was performed by a DP74 colour camera (Olympus).

**Estimation of parameters**. To estimate the mean velocity ($v_0$), the mean rotation rate ($\omega_0$), the standard deviation of rotation rate ($\sigma_\omega$), and the correlation time of rotation rate ($\tau$), the behaviours of sparsely isolated worms on the inner surface of the lid of a Petri plate were recorded using Olympus cellSens software and an SZX7 microscope equipped with the DP74 camera. From the obtained images, the centroid of 38 worms at 60 consecutive time points with an interval $\Delta t = 0.21669$ s were measured. $v_0$ was estimated from the average of the distance between two consecutive positions. Let the position of the $i$-th worm at $t$ be $(x^i(t), y^i(t))$. The direction of motion, $\theta^i(t)$, was calculated as the direction of $(x^i(t + \Delta t) - x^i(t), y^i(t + \Delta t) - y^i(t))$. Then, $\omega_0$ was estimated as $\langle \theta^i(t_e) - \theta^i(0) \rangle / t_e$, where $t_e = 59 \times \Delta t$, and $\langle \cdot \rangle$ means the average over $i$. Here, $\tau$ is the relaxation time of the change rate of $\theta^i(t)$ to $\omega_0$. If $\theta^i(t) = \int dt \left( \omega_0 + (\omega^i_{t=0} - \omega_0) e^{-t/\tau} \right) = \theta^i_0 + \omega_0 t + (\omega^i_{t=0} - \omega_0) \tau (1 - e^{-t/\tau})$,

$$\left\langle \frac{\theta^i(t_e) - \theta^i(0) - \omega_0 t_e}{|\theta^i(t_e) - \theta^i(0) - \omega_0 t_e|} (\theta^i(t) - \omega_0 t) \right\rangle$$
$$= \left\langle \frac{\theta^i(t_e) - \theta^i(0) - \omega_0 t_e}{|\theta^i(t_e) - \theta^i(0) - \omega_0 t_e|} \theta^i_0 \right\rangle + \left\langle |\omega^i_{t=0} - \omega_0| \right\rangle \tau \left(1 - e^{-\frac{t}{\tau}}\right) \quad (6)$$
$$= \left\langle \frac{\theta^i(t_e) - \theta^i(0) - \omega_0 t_e}{|\theta^i(t_e) - \theta^i(0) - \omega_0 t_e|} \theta^i_0 \right\rangle + \sqrt{\frac{2}{\pi}} \sqrt{\left\langle (\omega^i_{t=0} - \omega_0)^2 \right\rangle} \tau \left(1 - e^{-\frac{t}{\tau}}\right)$$

on the assumption that $\omega^i_{t=0} - \omega_0$ were normally distributed and $(\theta^i(t_e) - \theta^i(0) - \omega_0 t_e)(\omega^i_{t=0} - \omega_0) > 0$. $\tau$ and $\sigma_\omega$ were estimated from the fitting of $\left\langle \frac{\theta^i(t_e) - \theta^i(0) - \omega_0 t_e}{|\theta^i(t_e) - \theta^i(0) - \omega_0 t_e|} (\theta^i(t) - \omega_0 t) \right\rangle$ to $\theta_0 + \sqrt{2/\pi} \sigma_\omega \tau (1 - e^{-\frac{t}{\tau}}) \approx \theta_0 + \sqrt{2/\pi} \sigma_\omega (t - \frac{t^2}{2\tau})$.

**Measurement of the size of areas surrounded by bundles**. To measure the area sizes, the moving averaged images of collective motion were first binarized using Fiji (https://fiji.sc). The window width was 50 s in the *C. elegans* case and 100 time units in the model case. The binarized images were skeletonized using Matlab. The

sizes of the areas surrounded by bundles were measured by counting the pixels surrounded by lines in the skeletonized images.

## Data availability

All data supporting the findings in this study are available from the corresponding authors on reasonable request.

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

## Acknowledgements

The authors thank the *Caenorhabditis* Genetics Center for sharing strains, E. Okumura for initial technical assistance, T. Ishihara, M. Fujiwara (Kyushu University), T. Naka-gaki, K. Sato and S. Ochi (Hokkaido University) for valuable comments on this manu-script and discussion. This work was supported by JSPS KAKENHI Grant-in-Aid for Scientific research (B) (Grant no. JP17KT0016), on Innovative areas "Science of Soft Robot" project (Grant no. JP18H05474), and for Young Scientists (B) (Grant no. JP17K17761). T.S. was supported by the Japan Science and Technology Agency under Precursory Research for Embryonic Science and Technology (PRESTO) (Grant no. JPMJPR12A9) and by the Mochida Memorial Foundation for Medical and Phar-maceutical Research.

## Author contributions

H.I. discovered the phenomenon of dynamical network formation by *C. elegans* with the help of T.S. T.S., H.I. and K.H.N. conceived the experiments, and T.S. performed the experiments with M.N. K.H.N. conceived and analysed the simulations. T.S., H.I. and K.H.N. analysed the obtained images and wrote the paper.

## Additional information

**Competing interests:** The authors declare no competing interests.

