## [Peer Review File · Nature Communications]

Reviewers' comments:

Reviewer #1 (Remarks to the Author):

The paper reports extensive experimental work on collective behavior of a large population of *C. elegans* on 2D surfaces. Under a broad range of experimental conditions, an evolving dynamical network of active worms was observed. It was shown in particular that humidity or moisture inside the experimental setup has strong effect on the size of the compartments and the thickness of the worm bundles that delineate them. The authors then investigated a simple model of active particles in the literature, and obtained network structures that statistically resemble the ones seen in the worm experiments.

To my knowledge, the experimental system is new and offers an exciting addition to the repertoire of active systems that can be studied under lab conditions. The collective behavior discovered is interesting on its own. Therefore I am generally positive about publishing the work in *Nature Communications*. However, the model is not well-motivated in the current context and its study does not add much to the understanding of the pattern-forming process in the experimental system. Along this line, I found the last two sentences in the abstract to be excessive over-statements in terms of what we can learn about information processing using concepts from active matter physics. Additional comments follow.

1) It appears that the authors are well aware of the effect of water condensation on the size of worm bundle and compartment size (I suspect the two are closely related). Furthermore, activity of the worms not only generates flow within the bundle network, but also sideways streaming into the compartments. I would like to see the active matter physics being examined from such a perspective.

2) In Fig. 4e, it seems that the compartment size in the simulations is mostly controlled by the mean turning frequency ω_0 instead of the strength of the "cohesive force", while the experiments show much stronger dependence on water availability (Fig. 2b). This again speaks for possibly different physical mechanisms at work in the experimental system and in the model, although they share certain global features.

Reviewer #2 (Remarks to the Author):

This paper describes the new and interesting observation that *C. elegans* forms complex, bundle-shaped aggregates under certain conditions (starvation, crowding, and a moist glass or plastic substrate). The authors then develop a biophysical model that explains how worms could form such aggregates, and show that this model is consistent with the effects of various extrinsic parameters (humidity, worm density) on aggregate formation and disaggregation. They also show that the effects of altering the manner of worm locomotion by mutation or optogenetics are also consistent with their model.

I think the basic observations in this paper are extremely interesting, and while I am not sure I am the best reviewer to evaluate the biophysical aspects of the authors' model, I thought the evaluation of its predictions based on variations in external parameters seemed sound. However, I think the genetic and optogenetic tests were a bit weak. Although I think an extensive analysis of more mutants is beyond the scope of this paper, I think a few additional controls are needed.

Specific comments:

1. The authors analyzed the *mec-4(e1611)* mutant as an example of a worm with higher curvature than wild-type. However, this is not the only parameter that is affected by that mutation--indeed, the defect of that mutant in mechanosensation might also have an effect on aggregation. And since the authors only looked at a single mutant allele, it is not even clear that the differences they observed relative to wild-type are related to *mec-4* (there are undoubtedly many other mutations in the strain since all worm mutants are derived from heavily mutagenized strains). I think the authors need to look at at least a couple more strains with altered curvature (possibly with mutations in different genes) to see if curvature is really that feature that correlates with differences in aggregation. I think they also should test another *mec-4* allele to see if loss of *mec-4* function itself is responsible for the phenotype they observe.

2. The authors' explanation of the optogenetic experiment, that optical activation of the touch neurons simply reduces the fraction of inactive animals, is also a big oversimplification. Activation of the touch neurons is known to first cause a reversal (backward locomotion) followed by a minute or so of increased forward locomotion. Presumably the direction of motion as well as speed would affect aggregation, but this is not taken into account as far as I can tell. I don't have a good suggestion for an alternative experiment, but I think the authors at least need to acknowledge and explain the limitations to their experimental interpretation.

Reviewer #3 (Remarks to the Author):

This is a well-written paper about a very interesting system. The information is clear and concise, the work is well-referenced and the system under study is astonishing. This work combines experimental observations about the swarming of colonies of the nematode *C.elegans* with a mathematical model that correctly captures the main aspects of the swarm dynamics. This is Systems Science at its very best. I can only congratulate the authors for this excellent work.

I have one observation I suggest the authors address before the manuscript is published.

Towards the end of the main text (lines 191 to 196), the authors mention that the model also reproduces the particle light activation experiments and refer the reader to Movie S11. However, from this movie, I do not see that the same behavior is observed in the experiments (movies S6 and S7) and in the model (movie S11). I suggest the authors provide a more quantitative analysis to compare the disintegration of the clusters with the activation of the inactive particles in both the experiments and in the model. Just with the naked eye, it is not possible to assess whether or not the model is correctly reproducing the behavior of the system when the inactive particles are activated.

I would like to congratulate again the authors for this excellent work.

Max Aldana.

Response to Reviewer #1:

Thank you for your kind comments. Following your comments, we changed the last two sentences in the abstract into the following sentences.

Lines 22-26: Namely, we revealed that there is a collective motion which can be described by a simple minimal model in a broad area of parameter space in groups of animals. Our results will pave the way for the understanding of biological functions of groups of animals via the concepts of active matter physics.

1) It appears that the authors are well aware of the effect of water condensation on the size of worm bundle and compartment size (I suspect the two are closely related). Furthermore, activity of the worms not only generates flow within the bundle network, but also sideway streaming into the compartments. I would like to see the active matter physics being examined from such a perspective.

As you pointed out in the comments 2) again, the dependence of compartment size on cohesive force is still elusive. Inside compartments, many particles also flew in the model case, although it's hard to recognize particles due to much lower concentration. Comparison of dependence of concentration inside compartments on the activity between model and experiment will clarify the role of cohesive force on bundle formation. Using the model, we failed in reproducing the streaming protruded from still aggregates.

Since careful investigation of what you pointed out will lead further understanding, we added the following sentences in the discussion.

Lines 244-248: In this study, individual trajectories in the network, and collapsing process and effects of reversal in the optical response are not investigated well. The careful investigation of these subjects will lead to further understanding of collective motion of *C. elegans*.

2) In Fig. 4e, it seems that the compartment size in the simulations is mostly controlled by the mean turning frequency ω_0 instead of the strength of the "cohesive force", while the experiments show much stronger dependence on water availability (Fig. 2b). This again speaks for possibly different physical mechanisms at work in the experimental system and in the model, although they share certain global features.

We agree with this comment. Humidity dependence of compartment size was not very strong in our model. To make the limitation of our model clear, we added the following sentence.

Lines 198-199: It should be noted that the compartment size of real worms network more strongly depends on humidity than results from our model.

Lines 242-244: Although there are some of inconsistencies, e. g. compartment size dependence on humidity, we can state that our results provide a simple physical description of animal collective behaviours.

Response to Reviewer #2:

Thank you for your kind comments and useful suggestions about genetic experiments. We are very proud to hear “**extremely interesting**” from you.

1. The authors analyzed the *mec-4(e1611)* mutant as an example of a worm with higher curvature than wild-type. However, this is not the only parameter that is affected by that mutation--indeed, the defect of that mutant in mechanosensation might also have an effect on aggregation. And since the authors only looked at a single mutant allele, it is not even clear that the differences they observed relative to wild-type are related to *mec-4* (there are undoubtedly many other mutations in the strain since all worm mutants are derived from heavily mutagenized strains). I think the authors need to look at at least a couple more strains with altered curvature (possibly with mutations in different genes) to see if curvature is really that feature that correlates with differences in aggregation. I think they also should test another *mec-4* allele to see if loss of *mec-4* function itself is responsible for the phenotype they observe.

We first examined two strains, *snt-1(md290)* and *let-60(lf)* mutants, both of which have been known to move with higher curvature (Krajacic et al. Genetics, 2012; Hamakawa et al. BMC Biol, 2015). However, these mutants did not show enough propagation in dog food agar medium. According to your suggestion, we next tried to examine the network formation of the other *mec-4* alleles, *mec-4(e1339)* and *mec-4(u253)*. It was shown that the *mec-4(u253)* mutant hardly climbed up to the lid of the NGM plate, whereas the *mec-4(e1339)* mutants formed network on the lid like the *mec-4(e1611)* mutant. We confirmed that compartment sizes of the network in this mutant are clearly smaller than those in the wild-type worm and similar to that in *mec-4(e1611)* mutant. Because this result supports that loss of MEC-4 function itself is responsible for the observed phenotype, we added the new figure in Extended Data Figure 2 and the description about the result of the *mec-4(e1339)* in lines 103-106. On the other hand, since the possibility that another defect in the mutants, such as mechanosensation defect, might also have an effect on aggregation could not be excluded as of now, we mentioned this possibility in lines 106-111.

2. The authors' explanation of the optogenetic experiment, that optical activation of the touch neurons simply reduces the fraction of inactive animals, is also a big oversimplification. Activation of the touch neurons is known to first cause a reversal (backward locomotion) followed by a minute or so of increased forward locomotion. Presumably the direction of motion as well as speed would affect aggregation, but this is not taken into account as far as I can tell. I don't have a good suggestion for an alternative experiment, but I think the authors at least need to acknowledge and explain the limitations to their experimental interpretation.

Following your comment, we modified the explanation of optical stimulation in lines 114-123, and the following sentences were added.

Lines 210-212: Although light stimulation leads to both reversal and forward acceleration of worms, at least we can conclude that decrease of halted worms is one of the main causes of the instability of the bundles.

Lines 244-248: In this study, individual trajectories in the network, and collapsing process and effects of reversal in the optical response are not investigated well. The careful investigation of these subjects will lead to further understanding of collective motion of *C. elegans*.

Response to Reviewer #3:

Thank you very much for your positive comments. Your comments encourage us to study the system further.

Towards the end of the main text (lines 191 to 196), the authors mention that the model also reproduces the particle light activation experiments and refer the reader to Movie S11. However, from this movie, I do not see that the same behavior is observed in the experiments (movies S6 and S7) and in the model (movie S11). I suggest the authors provide a more quantitative analysis to compare the disintegration of the clusters with the activation of the inactive particles in both the experiments and in the model. Just with the naked eye, it is not possible to assess whether or not the model is correctly reproducing the behavior of the system when the inactive particles are activated.

Indeed, we should compare the experiment and the model quantitatively. Since the bundles were branched after light stimulation, image analysis of the bundles is difficult and we don't succeed in quantifying the collapse. To clarify the present situation, we added the following sentences.

Lines 244-248: In this study, individual trajectories in the network, and collapsing process and effects of reversal in the optical response are not investigated well. The careful investigation of these subjects will lead to further understanding of collective motion of *C. elegans*.

Reviewer #1 :

The paper reports extensive experimental work on collective behavior of a large population of *C. elegans* on 2D surfaces. Under a broad range of experimental conditions, an evolving dynamical network of active worms was observed. It was shown in particular that humidity or moisture inside the experimental setup has strong effect on the size of the compartments and the thickness of the worm bundles that delineate them. The authors then investigated a simple model of active particles in the literature, and obtained network structures that statistically resemble the ones seen in the worm experiments.

To my knowledge, the experimental system is new and offers an exciting addition to the repertoire of active systems that can be studied under lab conditions. The collective behavior discovered is interesting on its own. Therefore I am generally positive about publishing the work in Nature Communications. However, the model is not well-motivated in the current context and its study does not add much to the understanding of the pattern-forming process in the experimental system. Along this line, I found the last two sentences in the abstract to be excessive over-statements in terms of what we can learn about information processing using concepts from active matter physics. Additional comments follow.

1) It appears that the authors are well aware of the effect of water condensation on the size of worm bundle and compartment size (I suspect the two are closely related). Furthermore, activity of the worms not only generates flow within the bundle network, but also sideways streaming into the compartments. I would like to see the active matter physics being examined from such a perspective.

2) In Fig. 4e, it seems that the compartment size in the simulations is mostly controlled by the mean turning frequency ω_0 instead of the strength of the "cohesive force", while the experiments show much stronger dependence on water availability (Fig. 2b). This again speaks for possibly different physical mechanisms at work in the experimental system and in the model, although they share certain global features.

Reviewer #2 (Remarks to the Author):

This paper describes the new and interesting observation that *C. elegans* forms complex, bundle-shaped aggregates under certain conditions (starvation, crowding, and a moist glass or plastic substrate). The authors then develop a biophysical model that explains how worms could form such aggregates, and show that this model is consistent with the effects of various extrinsic parameters (humidity, worm density) on aggregate formation and disaggregation. They also show that the effects of altering the manner of worm locomotion by mutation or optogenetics are also consistent with their model.

I think the basic observations in this paper are extremely interesting, and while I am not sure I am the best reviewer to evaluate the biophysical aspects of the authors' model, I thought the evaluation of its predictions based on variations in external parameters seemed sound. However, I think the genetic and optogenetic tests were a bit weak. Although I think an extensive analysis of more mutants is beyond the scope of this paper, I think a few additional controls are needed.

Specific comments:

1. The authors analyzed the *mec-4(e1611)* mutant as an example of a worm with higher curvature than wild-type. However, this is not the only parameter that is affected by that mutation--indeed, the defect of that mutant in mechanosensation might also have an effect on aggregation. And since the authors only looked at a single mutant allele, it is not even clear that the differences they observed relative to wild-type are related to *mec-4* (there are undoubtedly many other mutations in the strain since all worm mutants are derived from heavily mutagenized strains). I think the authors need to look at at least a couple more strains with altered curvature (possibly with mutations in different genes) to see if curvature is really that feature that correlates with differences in aggregation. I think they also should test another *mec-4* allele to see if loss of *mec-4* function itself is responsible for the phenotype they observe.

2. The authors' explanation of the optogenetic experiment, that optical activation of the touch neurons simply reduces the fraction of inactive animals, is also a big oversimplification. Activation of the touch neurons is known to first cause a reversal (backward locomotion) followed by a minute or so of increased forward locomotion. Presumably the direction of motion as well as speed would affect aggregation, but this is not taken into account as far as I can tell. I don't have a good suggestion for an alternative experiment, but I think the authors at least need to acknowledge and explain the limitations to their experimental interpretation.

Reviewer #3 (Remarks to the Author):

This is a well-written paper about a very interesting system. The information is clear and concise, the work is well-referenced and the system under study is astonishing. This work combines experimental observations about the swarming of colonies of the nematode *C.elegans* with a mathematical model that correctly captures the main aspects of the swarm dynamics. This is Systems Science at its very best. I can only congratulate the authors for this excellent work.

I have one observation I suggest the authors address before the manuscript is published.

Towards the end of the main text (lines 191 to 196), the authors mention that the model also reproduces the particle light activation experiments and refer the reader to Movie S11. However, from this movie, I do not see that the same behavior is observed in the experiments (movies S6 and S7) and in the model (movie S11). I suggest the authors provide a more quantitative analysis to compare the disintegration of the clusters with the activation of the inactive particles in both the experiments and in the model. Just with the naked eye, it is not possible to assess whether or not the model is correctly reproducing the behavior of the system when the inactive particles are activated.

I would like to congratulate again the authors for this excellent work.

Max Aldana.

REVIEWERS' COMMENTS:

Reviewer #1 (Remarks to the Author):

While the authors confirmed observations and comments made in my previous report, they unfortunately did not seriously address the fundamentally different physical mechanisms that lead to seemingly similar network patterns. The sentence "Namely, we revealed that there is a collective motion which can be described by a simple minimal model in a broad area of parameter space in groups of animals" is still too strong a statement. Minimal model must include essential physics which in this case does not seem to be the lasting curly motion of worms assumed in the model but instead condensed water channels that serve to restrict homogenisation of the worm population.

Reviewer #2 (Remarks to the Author):

I think the authors have addressed the comments in the reviews, and I am supportive of publication.

Reviewer #3 (Remarks to the Author):

I think the authors have satisfactorily addressed my comment about this work. I understand that a mathematical model cannot reproduce ALL the phenomenology exhibited by a biological system as complicated as the one presented here. I also consider that the work presented here is an important step to understand a swarming phenomenon that has not previously been observed. The experiments are amazing and the mathematical model, although incomplete, does reproduce the main aspects of the system under consideration. Refinements and fine-tuning of the model should be implemented, but not in this article. The authors have explicitly stated the limitations of the model and that is enough for me. Therefore, I do not have more comments.

Maximino Aldana.

The followings are our responses to reviewers' comments and questions. We thank all the reviewers for their comments, which significantly improve our manuscript. All comments from the reviewers are attached in the end.

Response to Reviewer #1:

We would like to thank you for your valuable comments and also sincerely apologize for our insufficient previous revision. As you noted in the previous comments, we have also recognized possibly different physical mechanisms at work in the experimental system and in the model (although they share certain global features). According to your comments, we first removed the sentence " Namely, we revealed that there is a collective motion...in groups of animals." In addition, we changed the last sentence of Abstract to the following sentence which was kindly suggested by the Editor.

Our findings imply that the concepts of active matter physics may help us understand biological functions of animals groups.

We appreciate your constructive comments again and will try to the implementation of refinements and fine-tuning of the model as suggested by reviewer #3 in the future.

Reviewer #1 (Remarks to the Author):

While the authors confirmed observations and comments made in my previous report, they unfortunately did not seriously address the fundamentally different physical mechanisms that lead to seemingly similar network patterns. The sentence "Namely, we revealed that there is a collective motion which can be described by a simple minimal model in a broad area of parameter space in groups of animals" is still too strong a statement. Minimal model must include essential physics which in this case does not seem to be the lasting curly motion of worms assumed in the model but instead condensed water channels that serve to restrict homogenisation of the worm population.

Reviewer #2 (Remarks to the Author):

I think the authors have addressed the comments in the reviews, and I am supportive of publication.

Reviewer #3 (Remarks to the Author):

I think the authors have satisfactorily addressed my comment about this work. I understand that a mathematical model cannot reproduce ALL the phenomenology exhibited by a biological system as complicated as the one presented here. I also consider that the work presented here is an important step to understand a swarming phenomenon that has not previously been observed. The experiments are amazing and the mathematical model, although incomplete, does reproduce the main aspects of the system under consideration. Refinements and fine-tuning of the model should be implemented, but not in this article. The authors have explicitly stated the limitations of the model and that is enough for me. Therefore, I do not have more comments.

Maximino Aldana.